# Image retrieval outperforms diffusion models on data augmentation

**Max F. Burg**[1-3,*]**, Florian Wenzel**[4,*]**, Dominik Zietlow**[5,*]**, Max Horn**[6,*]**, Osama Makansi**[7]**, Francesco Locatello**[8,*]**, and Chris Russell**[9,*]

[1]International Max Planck Research School for Intelligent Systems, Tübingen, Germany
[2]Institute of Computer Science and Campus Institute Data Science, University of Göttingen, Germany
[3]Tübingen AI Center, University of Tübingen, Germany
[4]Mirelo AI
[5]ELLIS Institute Tübingen, Germany
[6]GlaxoSmithKline, AI&ML, Zug, Switzerland
[7]Amazon, Tübingen, Germany
[8]Institute of Science and Technology Austria
[9]Oxford Internet Institute, University of Oxford, United Kingdom

[*]Work mostly done at Amazon Web Services, Tübingen, Germany
 Contact: max.burg@bethgelab.org

**Reviewed on OpenReview:** `https://openreview.net/forum?id=xflYdGZMpv`

## Abstract

Many approaches have been proposed to use diffusion models to augment training datasets for downstream tasks, such as classification. However, diffusion models are themselves trained on large datasets, often with noisy annotations, and it remains an open question to which extent these models contribute to downstream classification performance. In particular, it remains unclear if they generalize enough to improve over directly using the additional data of their pre-training process for augmentation. We systematically evaluate a range of existing methods to generate images from diffusion models and study new extensions to assess their benefit for data augmentation. Personalizing diffusion models towards the target data outperforms simpler prompting strategies. However, using the pre-training data of the diffusion model alone, via a simple nearest-neighbor retrieval procedure, leads to even stronger downstream performance. Our study explores the potential of diffusion models in generating new training data, and surprisingly finds that these sophisticated models are not yet able to beat a simple and strong image retrieval baseline on simple downstream vision tasks.

## 1 Introduction

Data augmentation is a key component of training robust and high-performing computer vision models, and it has become increasingly sophisticated: From the early simple image transformations (random cropping, flipping, color jittering, and shearing) (Cubuk et al., 2020), over augmenting additional training data by combining pairs of images, such as MixUp (Zhang et al., 2017) and CutMix (Yun et al., 2019), all the way to image augmentations using generative models. Augmentation via image transformations improves robustness towards distortions that resemble the transformation (Wenzel et al., 2022b) and interpolating augmentations are particularly helpful in situations where diverse training data is scarce (Ghiasi et al., 2021). With the success of generative adversarial networks (GANs), generative models finally scaled to high-dimensional domains and allowed the generation of photorealistic images. The idea of using them for data augmentation purposes has been prevalent since their early successes and forms the basis of a new set of data augmentation

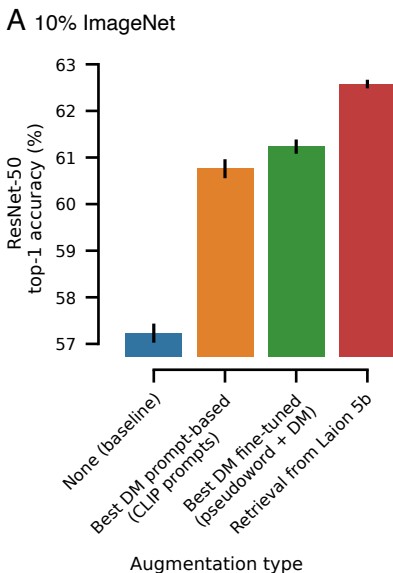
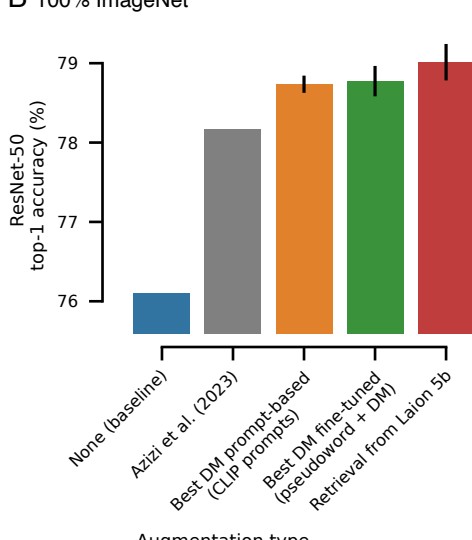

Figure 1: *Are generative methods beneficial for data augmentation?* **A.** Each bar shows the accuracy and standard error of the downstream classifier (ResNet-50, He et al. (2016)) based on the best augmentation method within each family on the data-scarce 10% sub-sample of ImageNet. While diffusion model (DM) based techniques (orange: a prompting method based on Sariyildiz et al. (2022); and green: a fine-tuned DM) improve over the baseline of only using the original data (blue), a simple, more computationally efficient retrieval method using images from the diffusion model's pre-training dataset directly (red) performs best. **B.** Methods found to be best on 10% ImageNet perform on full ImageNet as well. All methods suggested in this paper outperform Azizi et al. (2023) in a like-with-like comparison that matches data and classifier architecture (gray; ResNet-50 performance reported in their Table 3, column "Real+Generated").

strategies (Zietlow et al., 2022; Ghosh et al., 2022; Antoniou et al., 2017; Frid-Adar et al., 2018; Esteban et al., 2017; Motamed et al., 2021; Mariani et al., 2018; Ramaswamy et al., 2021).

Given the emergence of diffusion models (DMs) that outperform GANs in terms of visual quality and diversity (Ramesh et al., 2022; Saharia et al., 2022b; Rombach et al., 2022), using them for data augmentation is a natural next step. Diffusion models are trained on a large dataset of billions of filtered image text pairs retrieved from the internet, enabling them to generate images of unparalleled variety. Furthermore, these models can be easily conditioned using text queries, which allows for easy control of the data generation process.

We benchmark a variety of existing augmentation techniques based on stable diffusion models (Rombach et al., 2022) on a randomly sampled 10% subset of the ImageNet Large Scale Visual Recognition Challenge 2012 (ILSVRC2012) (Russakovsky et al., 2015), simulating a low-data regime because adding data through augmentation helps most when training data is scarce (Hyams et al., 2019). Surprisingly, we find that these techniques are outperformed by simply retrieving nearest neighbors from the DM's training dataset (Laion 5b; Schuhmann et al. (2022)) with simple prompts (using the same CLIP-like model (Radford et al., 2021) used in the DM). We propose new extensions and variants of diffusion-model-based strategies each leading to improvements, however, none beat our simple retrieval baseline. We show that simple text prompts based on class labels suffice for conditioning the DMs to improve the performance of standard classifiers (we chose ResNet-50; He et al. (2016)) compared to the unaugmented dataset. As images generated by simple prompts match the training distribution of the DM and not the training distribution of the classifiers, we test—inspired by related work on personalized DMs—methods that fine-tune the DM conditioning and optionally the DMs denoising model component. These fine-tuned models outperformed the best prompting strategies for their ability to create even better synthetic data for augmentation on ImageNet (Figure 1). Furthermore, for the best prompt-based and fine-tuned DM, we verified results obtained on 10% ImageNet (Figure 1A) generalize

to full ImageNet (Figure 1B) outperforming Azizi et al. (2023), and found similar results on Caltech256 (Figure 6; Griffin et al. (2007)). Here, augmentation leads to smaller performance boosts compared to the data-scarce setting, which is expected since abundant training data generally makes augmentation less important (Hyams et al., 2019). Although all investigated methods based on generating synthetic images were sophisticated and compute-intensive, none of them outperformed simple nearest neighbor retrieval (Figures 1 and 6).

As the Laion 5b dataset provides a compute-optimized search index to retrieve images which are most similar to a text prompt, our retrieval approach is computationally efficient (only training the classifier requires a GPU). Importantly, retrieval does not require pre-downloading the entire dataset, rather, only storing the small search index and downloading nearest neighbor images referenced by the index is necessary. As such, even where the performance improvement is minor over diffusion model based augmentation methods (Figures 1B and 6), we believe that the conceptual simplicity and efficiency of our approach makes it a compelling alternative.

## 2 Related work

**Data augmentation using generative adversarial networks.** Data augmentation is widely used when training deep networks. It overcomes some challenges associated with training on small datasets and improving the generalization of the trained models (Zhang et al., 2017). Manually designed augmentation methods have limited flexibility, and the idea of using ML-generated data for training has attracted attention. Generative adversarial networks (GAN) such as BigGAN (Brock et al., 2018) have been used to synthesize images for ImageNet classes (Besnier et al., 2020; Li et al., 2022). Despite early promising results, the use of GANs to generate synthetic training data has shown limited advantages over traditional data augmentation methods (Zietlow et al., 2022). Diffusion models, on the other hand, might be a better candidate since they are more flexible via general-purpose text-conditioning and exhibit a larger diversity and better image quality (Sohl-Dickstein et al., 2015; Ramesh et al., 2022; Saharia et al., 2022b; Rombach et al., 2022). Hence, in this work, we focus on diffusion models.

**Data augmentation using diffusion models.** Recently, diffusion models showed astonishing results for synthesizing images (Sohl-Dickstein et al., 2015; Ramesh et al., 2022; Saharia et al., 2022b; Rombach et al., 2022). Numerous approaches have been published that adapt diffusion models to better fit new images and that can be used for augmentation. We evaluate methods that employ diffusion models in a guidance-free manner or prompt them without adapting the model: Luzi et al. (2022) generate variations of a given dataset by first adding noise to the images and then denoising them again, and Sariyildiz et al. (2022) generate a synthetic ImageNet clone only using the class names of the target dataset. We also evaluate methods for specializing (AKA personalizing) diffusion models in our study. Given a few images of the same object (or concept), Gal et al. (2022) learn a joint word embedding (pseudoword) that reflects the subject and can be used to synthesize new variations of it (e.g., in different styles) and Gal et al. (2023) recently extended their method to significantly reduce the number of required training steps. Kawar et al. (2022) follow a similar approach and propose a method for text-conditioned image editing by fine-tuning the diffusion model and learning a new word embedding that aligns with the input image and the target text. We investigate the usefulness of "personalization" for data augmentation, which was not conclusively addressed in the original papers. We propose and evaluate extensions of these methods tailored to data augmentation and provide a thorough evaluation in a unified setting. Finally, we benchmark these approaches against our suggested retrieval baseline.

There have been multiple concurrently proposed methods to synthesize images for various downstream tasks that we could not include in our evaluation of their helpfulness for data augmentation to train an image classifier. Some methods focus on fine-tuning the diffusion model learning a unique identifier to personalize the DM to the given subject (Ruiz et al., 2022), Shipard et al. (2023) create synthetic clones of CIFAR (Krizhevsky et al., 2009) and EuroSAT (Helber et al., 2019) for zero-shot classification, other methods employ alternative losses for image generation to improve few-shot learning (Roy et al., 2022), and Packhäuser et al. (2022); Ghalebikesabi et al. (2023) create synthetic, privacy-preserving clones of medical data. Other methods optimize the features of the embedded images to augment specifically small datasets

(Zhang et al., 2022). Trabucco et al. (2023) additionally employ image synthesis, image editing (Meng et al., 2021) and in painting (Lugmayr et al., 2022; Saharia et al., 2022a), Bansal & Grover (2023) generate new images conditioning the DM by prompts and example images leading to degraded performance when used to augment ImageNet, and other work focuses on medical data and sample curation (Akrout et al., 2023).

He et al. (2023) use pretrained CLIP (Radford et al., 2021) as a base classifier and fine-tune it on synthetic images to better match the desired target distribution (e.g. ImageNet). Their approach improved zero- and few-shot classification performance. To generate the images, they sampled from a frozen pretrained DM with sophisticated class-name-based guidance and filtered them to keep only synthetic images with high CLIP classification confidence. In the few-shot case, they used real images to initialize image generation and to remove generated images showing high classification probabilities for wrong classes. Our work does not address improving an already performing, pretrained CLIP model. Rather, we compared ResNet-50 classifier performance trained from scratch on datasets augmented by synthetic images, which we generated from pretrained and fine-tuned diffusion models, and by images retrieved from a web-scale dataset. Most similar to our study is the concurrent work of Azizi et al. (2023), augmenting ImageNet with samples generated by a fine-tuned Imagen diffusion model (Saharia et al., 2022b). For fine-tuning to ImageNet, they trained parameters of the generative DM while excluding the class-name-based conditioning from optimization. In comparison, all of our augmentation methods performed better (Figure 1B). Most importantly, none of the studies mentioned in this section compared against our suggested retrieval baseline.

## 3 Experimental protocol

We consider a wide range of generative and retrieval-based augmentation techniques and evaluate their performance on a downstream classification task (Figure 2A).

**Dataset.** *ImageNet.* To extensively benchmark augmentation techniques, we simulate training data in a low-data regime, as generally, augmentation is most helpful when training data is scarce (Hyams et al., 2019). Hence, unless explicitly stated, we use a 10% sample (126,861 training images) of the ImageNet Large Scale Visual Recognition Challenge 2012 (ILSVRC2012) (Russakovsky et al., 2015) training split retaining class imbalance throughout our experiments. As the retrieval method did not return sufficient samples for 10 of the 1,000 ImageNet classes, they were excluded from augmentation, model training, and evaluation. We additionally sample a disjoint set of images of the same size from the training split for hyperparameter optimization and model selection, which we refer to as the validation split. When evaluating if results found on the 10% sub-sample generalize to full ImageNet, we include all but our validation split to the training set. All trained classification models are evaluated on the original ILSVRC2012 validation split. *Caltech256.* We additionally verify our findings on Caltech256 (Griffin et al., 2007) excluding five of the 256 classes for which retrieval did not return sufficient samples. We randomly sampled 80% of the data as train split and randomly partitioned the remaining 20% of the samples into equally sized disjoint validation and test splits.

**Diffusion model backbone.** To generate images, we used the pretrained Stable Diffusion v1.4 model, based on a latent diffusion architecture (Rombach et al., 2022). We discarded images marked as NSFW by the provided safety checker, replacing them with new samples. Stable Diffusion was trained on image sizes of 512 px, and we kept this resolution for all methods.

**Data augmentation and classifier training.** Diffusion models stochastically generate images, leading to a distribution of images that we wanted to capture when evaluating classifier performance. Hence, for each DM-based augmentation strategy, we generate 390 images per class ensuring that the number of images at least tripled per class, as in our sub-sampled ImageNet dataset each class contains between 74 and 130 images. For full ImageNet, we generate ten times more images. Then, we resample that data into five sets containing the same number of samples per class as our target dataset. We add each of the five additional datasets to the original training data for augmentation, obtaining five augmented datasets that we use for training a ResNet-50 classifier (He et al., 2016) on each of them to derive variance estimates. On Caltech256 we generate three times as many images per class as in the training set and follow the same approach as for ImageNet. During training, the samples are further augmented by random resizing and cropping as is

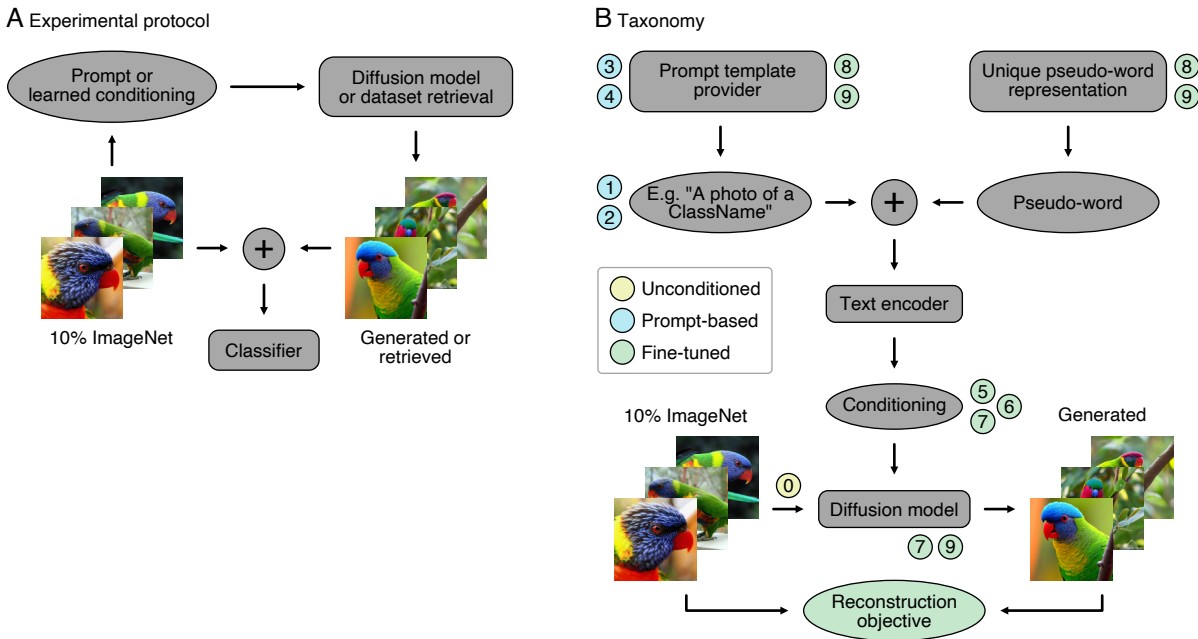

Figure 2: *Experimental protocol and taxonomy of diffusion model based augmentation methods.* **A.** We generate images by guiding the Stable Diffusion model by text prompts or by learned conditioning, or retrieving images by a nearest neighbor search in the CLIP embedding space of the DM's training data. We then train the downstream classifier on the original 10% ImageNet data augmented by the additional data and evaluate on the original ImageNet validation split. **B.** All considered methods adapt different components of the prompting, conditioning mechanisms, and the fine-tuning of the diffusion model. We reference each method by a circled number (see section 3.1 for details). Some methods edit the prompts while keeping the DM frozen: ①, ② use a single prompt for each class and ③, ④ use multiple prompts from a set of templates. Another family of methods optimize the conditioning vectors for the given images: ⑤, ⑥ only optimize the conditioning vector keeping the DM frozen, while ⑦ also jointly fine-tunes the DM. Instead of optimizing the conditioning vector, ⑧ learns a pseudo-word description of the class using multiple prompts keeping the DM frozen, while ⑨ additionally fine-tunes the DM. ⓪ does not adapt any component of the DM and relies on encoding and decoding to create variations of the given images.

common for classifier training on ImageNet. On ImageNet we trained the classifier with a batch size of 256 and a learning rate of 0.1. On Caltech256 we used a batch size of 1024 and after an initial learning rate optimization sweep we set the learning rate to 0.003. We divided the learning rate by 10 when validation accuracy failed to improve for 10 epochs. We stopped training when the validation accuracy did not improve for 20 epochs or after at most 270 epochs and used the highest validation accuracy checkpoint for final scoring.

## 3.1 Augmentation methods

The benchmarked augmentation methods can be grouped into four categories: (1) guidance-free diffusion model sampling, (2) simple conditioning techniques with prompts based on the objects' class label, and (3) personalization techniques that fine-tune the diffusion model conditioning and optionally the diffusion model itself to the classifier's data domain. We compare diffusion model approaches to a simple baseline (4) using images retrieved from the dataset that the diffusion model was trained on.

**Unconditional generation.** Following the procedure of ⓪ BOOMERANG (Luzi et al., 2022), we investigated a guidance-free method that does not require conditioning or updating the diffusion model. Instead, the approach adds noise to individual samples before denoising them.

**Prompt conditioning.** We explore several prompt-based methods of guiding the DM to produce samples for a specific class (Figure 2B). ① SIMPLE PROMPT: We condition the model by simple prompts containing the object's class name $n$, prompting the DM with "A photo of $n$." (method proposed in this paper) and a version of it, ② SIMPLE PROMPT (NO WS), stripping whitespace, $w(\cdot)$, from class names, "A photo of $w(n)$." (proposed). ③ CLIP PROMPTS: We add sampling prompts from the set of CLIP (Radford et al., 2021) text-encoder templates, e.g. "a photo of many $w(n)$.", "a black and white photo of the $w(n)$.", etc. (proposed) and ④ SARIYILDIZ ET AL. PROMPTS: a set of templates proposed to create a synthetic ImageNet clone (Sariyildiz et al., 2022).

To specify the class name, we leverage ImageNet's class name definitions provided by WordNet (Miller, 1995) synsets representing distinct entities in the WordNet graph. Each synset consists of one or multiple lemmas describing the class, where each lemma can consist of multiple words, e.g., "Tiger shark, Galeocerdo Cuvieri". We link each class via its synset to its class name. If a synset consists of multiple lemmas, we separate them by a comma, resulting in prompts like "A photo of tiger shark, Galeocerdo Cuvieri.", as we found that providing multiple lemmas led to better performance than using only the first lemma of a synset. Whenever methods inserted the class name into prompt templates, we sampled the templates randomly with replacement. Sariyildiz et al. (Sariyildiz et al., 2022) provided multiple categories of prompt templates (e.g. class name only, class name with WordNet hypernyms, additionally combined with "multiple" and "multiple different" specifications, class name with WordNet definition, and class name with hypernyms and randomly sampled backgrounds from the places dataset (Zhou et al., 2017)). Here, we sampled for each category the same number of images and randomly across the background templates. For Caltech256 we used the class names as provided by the dataset.

**Fine-tuning the diffusion model.** We explore various methods for fine-tuning a DM for class-personalized sampling to improve reconstruction of the classification dataset (Figure 2B). ⑤ FT CONDITIONING: freezing the DM and optimizing one conditioning per class (method proposed in this paper). ⑥ FT CLUSTER CONDITIONING: optimizing multiple conditionings per class rather than just one (proposed). ⑦ inspired by IMAGIC (Kawar et al., 2022), jointly fine-tuning the conditioning and the DM's denoising component. ⑧ TEXTUAL INVERSION (Gal et al., 2022): instead of fine-tuning the conditioning, sampling prompt templates and combining them with optimizing a pseudo-word representing the class-concept. ⑨ PSEUDOWORD+DM: combining the previous approach with optimizing the DM's denoising component.

We trained all models with the default Stable Diffusion optimization objective (Rombach et al., 2022) until the validation loss stagnated or increased – no model was trained for more than 40 epochs. When fine-tuning the diffusion models on full ImageNet, the number of training samples increased 10-fold, thus, to keep the computational budget comparable across modalities, we stopped training after 3 epochs. We set hyperparameters in accordance with published works (Gal et al., 2022; Kawar et al., 2022; Luzi et al., 2022) or existing code where available. Scaling to larger batch sizes was implemented by square-root scaling the learning rate to ensure constant gradient variance, and we performed a fine-grained grid-search for the optimal learning rate. For the FT CLUSTER CONDITIONING models, we clustered the training images within a class using k-means on the inception v3 embeddings. The conditioning vectors were initialized by encoding the SIMPLE PROMPT (NO WS) prompts with the DM's text encoder. For textual inversion (Gal et al., 2022), we fine-tuned the text embedding vector corresponding to the introduced pseudoword and sample from the provided text templates (Gal et al., 2022) to optimize the reconstruction objective for our ImageNet subset, freezing all other model components. We initialized the pseudo-word embedding with the final word in the first synset lemma (i.e., for the class "tiger shark" we used "shark"). Where this initialization resulted in multiple initial tokens, we initialized with the mean of the embeddings. For each fine-tuning run we stored checkpoints with the best train and validation loss and those corresponding to 1, 2 and 3 epochs of training. In the case of IMAGIC (Kawar et al., 2022) we found that the validation loss was still decreasing after 40 epochs, however, as the quality of the reconstructed images significantly deteriorated after 8 epochs, we instead stored checkpoints for the first 8 epochs. This is similar to the Imagic training scheme, which optimized the embedding for 100 steps and the U-Net for 1,500 steps on a single image. Although we follow existing procedure as closely as possible, image quality might improve for longer model training runs. We selected the final model checkpoint based on the validation accuracy of a ResNet-50 classifier trained on the union of augmented samples and the target dataset.

Table 1: *Overall performance of the data augmentation methods.* We report the top-1 accuracy and standard error of the downstream ResNet-50 classifier (He et al., 2016) trained on the augmented 10% ImageNet subset. The methods are described in section 3.1 and are grouped into families as described in the beginning of section 4. The circles refer to the taxonomy in Figure 2B.

| Augmentation method | Accuracy (%) |
|---|---|
| **10% ImageNet** | **57.2 ± 0.2** |
| **20% ImageNet** | **70.2 ± 0.3** |
| **Boomerang** (Luzi et al., 2022) ⓪ | **56.3 ± 0.3** |
| Simple prompt (no ws; proposed) ② | 60.0 ± 0.3 |
| Simple prompt (proposed) ① | 60.1 ± 0.2 |
| Sariyildiz et al. prompts (Sariyildiz et al., 2022) ④ | 60.8 ± 0.2 |
| **CLIP prompts** (proposed) ③ | **60.9 ± 0.2** |
| FT conditioning (proposed) ⑤ | 60.8 ± 0.1 |
| FT cluster conditioning (proposed) ⑥ | 60.9 ± 0.2 |
| Imagic (conditioning & DM; Kawar et al. (2022)) ⑦ | 61.0 ± 0.3 |
| Textual inversion (pseudoword; Gal et al. (2022)) ⑧ | 61.0 ± 0.4 |
| **Pseudoword+DM** (combining Gal et al. (2022); Kawar et al. (2022)) ⑨ | **61.2 ± 0.2** |
| **Retrieval (our suggested baseline)** ⑩ | **62.6 ± 0.1** |

**Laion nearest neighbor retrieval.** As a baseline comparison, we suggest using ⑩ RETRIEVAL (method proposed in this paper) to select images from the Laion 5b (Schuhmann et al., 2022) dataset used to train the diffusion model. This method finds nearest neighbor images to the SIMPLE PROMPT (NO WS) class name prompts in the CLIP embedding space.

Laion 5b is a publicly available dataset of 5 billion image-caption pairs extracted via web-crawler. The data was then filtered by only retaining images where the CLIP image embedding was consistent with the caption embedding. This filtering acts as a weak form of supervision, that retains a set of images where CLIP is more likely to work. The dataset provides a CLIP embedding nearest neighbors search index for each instance, and an official package (Beaumont, 2022) allows for fast search and retrieval. We used this to retrieve 130 images per class for ImageNet and the same number of samples per class for Caltech256. Unlike generating images with DMs, retrieving images is not stochastic, and we augment with those retrieved samples that are most similar to the class name search query. Images with a Laion aesthetics score of less than 5 were discarded to allow a fair comparison with Stable Diffusion 1.4 which was trained on this subset. For a fair comparison to our generative augmentation methods, we used the same safety checker model to discard images that were marked as NSFW. Due to changes in the availability of the images at the URLs in the dataset and the described filtering steps, it is often necessary to retrieve more than the desired number of images, which we do by increasing the number of nearest neighbors gradually from $1.4 \cdot 130$ to $10 \cdot 1.4 \cdot 130$ when not enough samples were found. To avoid using the same image multiple times, we applied the duplicate detector of the clip-retrieval package (Beaumont, 2022), however, this does not detect all near duplicate images and some duplicates are still used.

**Retrieval is computationally efficient.** Just as simply prompting a pretrained diffusion model to generate images avoids the need to download and train on a large dataset like Laion 5b, the use of existing retrieval search indices allows us to only download those images closest to the search prompt, without downloading and indexing the entire dataset ourselves. This makes any retrieval component of the pipeline computationally efficient, and bottle necked by internet access rather than pure compute. As such, the performance numbers we report should be seen as representative, rather than being exactly replicable by someone using the same compute, because availability of images at the indexed internet addresses is not guaranteed.

Figure 3: *Example images generated by* BOOMERANG ([Luzi et al., 2022](#)). The generated images lack diversity and effectively only add noise to the original ones. **A.** Example images from 10% ImageNet. **B.** The augmentations produced by Boomerang. **C.** There is only a small difference between the original and augmented images (best viewed on screen).

We estimated compute requirements for all methods augmenting 10% ImageNet from log files and file time-stamps. For full ImageNet, requirements were approximately ten-fold higher. RETRIEVAL was the fastest method, only requiring a CPU machine with 4 GB of CPU RAM and a 1.6 TB SSD disk to store the search index, completing image retrieval in 28.5 h. Here, the bottleneck was downloading the images, which can be avoided if a local copy of Laion 5b's image caption pairs is available, as nowadays is the case for many institutions. Time-requirements for DM fine-tuning depended on the specific method: FT CLUSTER CONDITIONING: 244 GPU h, FT CONDITIONING: 380 GPU h, TEXTUAL INVERSION: 368 GPU h, IMAGIC: 552 GPU h, PSEUDOWORD+DM: 720 GPU h (trained on 8 Nvidia V100 GPUs using distributed data parallel training, i.e. 1 GPU h corresponds to $1/8$ h $= 7.5$ min training time on 8 parallel GPUs). Generating 390 images for each ImageNet class required 867 GPU h (Nvidia T4 GPU). Since ResNet-50 classifier training was early stopped, training times varied typically between 128 and 160 GPU h (trained on 8 Nvidia T4 GPUs using distributed data parallel training). Overall, RETRIEVAL was most efficient: 30x faster than the prompt-based DM methods, and 56x faster than the best performing and most compute intense PSEUDOWORD+DM, while not requiring an expensive GPU machine.

## 4 Results

We first analyze augmentation on the data-scarce 10% ImageNet subset, providing a results summary in Table 1 in which each block corresponds to the groups of augmentation methods introduced in Section 3.1. Section 4.1 introduces the baselines, section 4.2 benchmarks the previously suggested unconditional DM augmentation method and a method proposed to create a synthetic ImageNet clone ([Sariyildiz et al., 2022](#)) against our simple RETRIEVAL baseline method, section 4.3 discusses improvements of conditioning the DM with text prompts, and section 4.4 discusses personalization approaches to diffusion models. Finally, section 4.5 compares RETRIEVAL to synthetic augmentation data, explores scaling behavior, verifies that the methods in this paper generalize to full datasets, and certifies superior RETRIEVAL performance is not due to retrieving ImageNet duplicates for augmentation. We obtained all classification performance numbers by training a ResNet-50 ([He et al., 2016](#)).

### 4.1 Baselines

To establish the baseline classifier performance, we train on our 10% ImageNet subset (no added data), achieving a top-1 accuracy of $57.2 \pm 0.2$. Since all augmentation methods double the size of the dataset, we consider an upper bound performance by using 20% of original ImageNet (no added data) with a classifier accuracy of $70.2 \pm 0.3$.

### 4.2 Retrieval outperforms previously suggested diffusion-based augmentation

**Unconditional generation.** BOOMERANG ([Luzi et al., 2022](#)) generates augmented images by sequentially adding Gaussian noise and uses a diffusion model to denoise elements of the target dataset, without altering the diffusion model or prompts. This results in a lower top-1 accuracy of $56.3 \pm 0.3$ compared to the original subset of ImageNet. Figure 3 shows that this method does not induce significant diversity in the dataset

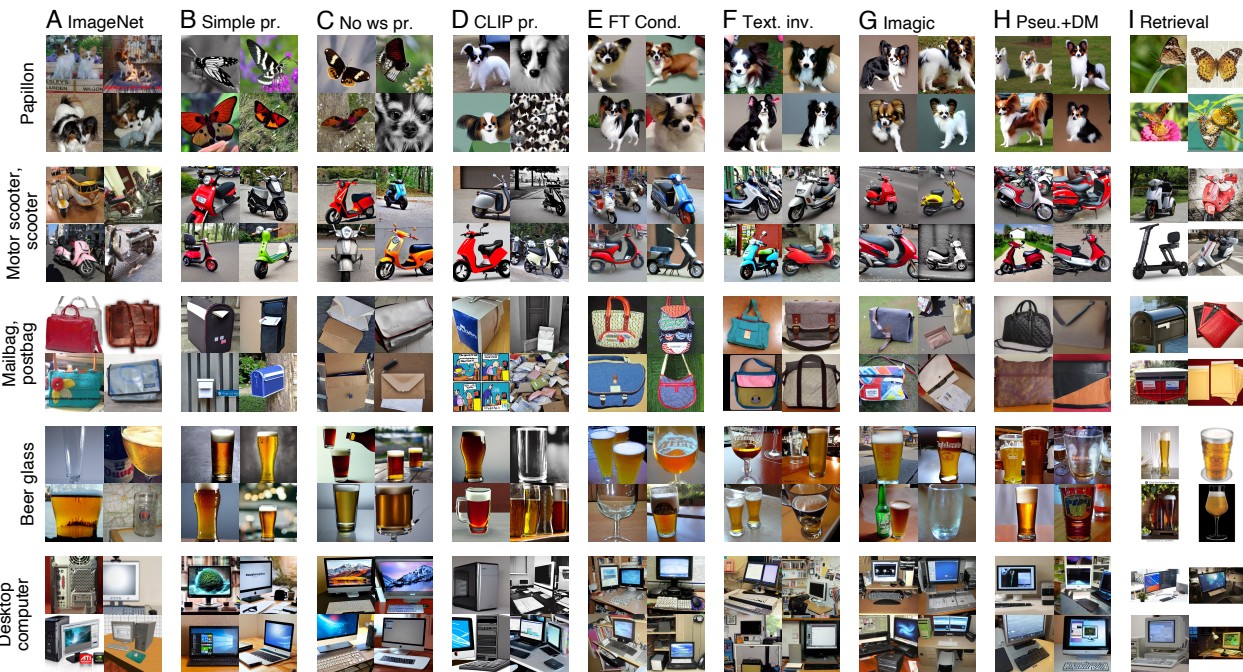

Figure 4: *Example images obtained from the investigated augmentation methods.* **A.** 10% ImageNet original images. **B.-D.** Images generated by our prompt-based sampling techniques: (B) SIMPLE PROMPT, (C) SIMPLE PROMPT (NO WS) and (D) CLIP PROMPTS, respectively, where in (D) we show images to the prompts "a bad photo of a $w(n)$", "a black and white photo of the $w(n)$", "a cartoon $w(n)$", and "a photo of many $w(n)$". **E.-H.** Images generated by the fine-tuned diffusion models, specifically (E) FT CONDITIONING, (F) TEXTUAL INVERSION, (G) IMAGIC, and (H) PSEUDOWORD+DM. **I.** Examples of RETRIEVAL from the diffusion model's training set. Best viewed when zoomed in.

and only slightly distorts the original images, leading to an overall decrease in performance. However, since this method is directly applied on the target dataset samples, it does not suffer from domain shift or class ambiguity.

**Prompt-conditioning.** Sariyildiz et al. (2022) generated augmented training examples with a diffusion model adapted to the target dataset via conditioning mechanisms. Their method guides image generation by various prompts based on the class name (see section 3.1 for details) and achieves $60.8 \pm 0.2$ classification accuracy, performing better than the 10% ImageNet baseline and worse than the 20% ImageNet upper bound (Table 1).

**Retrieving from the diffusion model pre-training data.** We now established that diffusion models are helpful for creating augmentation data, however, it is unclear how much value the DM's generative capabilities add compared to simply using their pre-training data directly for augmentation. To answer this question, we propose a simple RETRIEVAL method fetching images from the DM's pre-training data that are semantically closest to the SIMPLE PROMPT (NO WS) prompts (see section 3.1 for details). Augmenting with this data outperformed the sophisticated diffusion model based approaches at $62.6 \pm 0.1$ top-1 accuracy, while being computationally less demanding.

**Summary.** Using diffusion models off the shelf, i.e., without adaptation or conditioning using the target dataset, is not beneficial and can even deteriorate the classifier performance compared to the ImageNet baseline. Augmenting with generated samples adapted to the target dataset by prompt-conditioning performs better than the unaugmented baseline. However, simply augmenting with images retrieved from the diffusion model's training data outperforms the DM based approaches, indicating that DM generated images have

significant shortcomings. As retrieved images are real-world images, they show high photorealism, variety, and detail. However, retrieval underperforms the 20% ImageNet upper bound performance, which might be caused by a distribution shift in images between ImageNet and Laion 5b, and failures in retrieval, including both the retrieval of irrelevant images from the CLIP-based search index, and class ambiguity ("papillon" and "mailbag, postbag" in Figure 4I), reflecting a mismatch of concepts from CLIP latent space to ImageNet classes as we measured semantic similarity as distance of CLIP embeddings.

### 4.3 Advanced prompt-conditioning improves but is not competitive with retrieval

Then, we embarked on a quest asking if we can improve diffusion model based methods over the strong RETRIEVAL baseline. To do so, we first explored several variations of text prompt conditioning strategies.

**Simple prompt conditioning.** We investigate a SIMPLE PROMPT conditioning method that uses the prompt "A photo of $n$.", where $n$ is replaced by the class name, e.g., "A photo of tiger shark, Galeocerdo cuvieri." This improved over the 10% ImageNet baseline to a top-1 classification accuracy of $60.1 \pm 0.2$, but Figure 4B shows clear problems with class ambiguity and a lack of diversity.

**Tackling class ambiguity by white space removal.** While generally mitigating class ambiguity is a hard problem, we focus on the ambiguity introduced by class names composed of multiple words. To this end, we investigated the variant SIMPLE PROMPT (NO WS) (no white space) that removed the white space from the class name used by SIMPLE PROMPT, e.g., "A photo of desktopcomputer.". While class ambiguity reduced for some classes (e.g., desktop computer; Figure 4C) it was ineffective for others (e.g., papillon) and overall resulted in slightly degraded performance $60.0 \pm 0.3$ compared to keeping the white space. We explore more advanced methods in the following sections.

**Improving sample diversity by diverse prompt templates.** To increase the diversity of the samples, we made use of multiple prompt templates. We use CLIP PROMPTS which randomly selects one of the text templates provided by CLIP (Radford et al., 2021) (see Figure 4D for examples and Radford et al. (2021) for a full list), increasing classification accuracy to $60.9 \pm 0.2$. This method performed best among all prompt-based techniques. Interestingly, the overall performance increased even though some prompts lead to synthetic images that did not match the style of ImageNet samples (e.g., "a cartoon photo of the papillon.") or images with texture-like contents instead of objects (e.g., papillon, image to the bottom right in Figure 4D). Surprisingly, these slightly more elaborate templates just adding few more words compared to SIMPLE PROMPT (NO WS) (e.g., "a bad photo of a papillon.") improved class ambiguity in some cases (e.g., papillon; Figure 4D).

**Summary.** Augmenting a dataset with images sampled by prompt-based conditioning techniques improves the downstream classifier performance beyond the unaugmented datasets, however, none of these methods improve over RETRIEVAL. Inspecting the generated samples, we find that various challenges remain: ImageNet is more than a decade old, and some images included in the dataset are older and of different style when compared to the images created by the recently trained Stable Diffusion model (e.g., desktop computer; Figure 4A-D). In other cases, generated images do not match the domain of ImageNet samples, for instance because they are too artificial, sometimes even like a computer rendering (e.g., motor scooter; Figure 4A-D), their texture does not match (e.g., papillon; Figure 4A,D) or the prompt does not match the desired style (e.g., cartoon; Figure 4A,D). This tendency may have been amplified by Stable Diffusion v1.4 being trained on a subset of Laion, which was filtered to contain only aesthetic images.[1]

### 4.4 Personalizing the diffusion model improves further but does not outperform retrieval

In the previous section, we found that adapting the diffusion models by only editing the prompt offers limited improvement when used for augmentation. On our quest to improve diffusion models to narrow down the performance gap to the RETRIEVAL baseline, this section explores a more advanced set of methods that additionally fine-tune parts of the diffusion model to "personalize" it to the 10% ImageNet training images

---

[1] https://github.com/CompVis/stable-diffusion

by optimizing the DM reconstruction objective (c.f. Figure 2B). Originally, these methods were proposed in the context of personalizing the model to a specific object or concept, i.e., a single or only a few images.

**Fine-tuning the conditioning vectors.** The general diffusion model architecture has multiple components that can be fine-tuned (see Figure 2B). We start by fine-tuning one conditioning vector per class to optimize the reconstruction objective for our ImageNet subsample, while keeping the other model weights fixed. This method, dubbed FT CONDITIONING (fine-tune conditioning), achieves an augmentation accuracy of $60.8 \pm 0.1$ and is on par with the best prompt editing method CLIP PROMPTS. Interestingly, while this method achieves good augmentation performance, the generated images do not look photorealistic, and suffer from noise and missing backgrounds (Figure 4E).

**Fine-tuning clusters of conditioning vectors.** Using a single conditioning vector for all images from one class might be insufficient to capture the full class variability. To see if this is the case, we explore FT CLUSTER CONDITIONING. We generate $k$ clusters of images per class, before fitting a conditioning vector to each individual cluster (see section 3.1 for details). We find that using $k = 5$ clusters slightly improves the accuracy by 0.1 percentage points over using $k = 1$. For larger $k = 10$ and $k = 15$, performance decreased ($60.8 \pm 0.2$ and $60.6 \pm 0.3$ accuracy). This might be due to the smaller number of images per cluster, making the fine-tuning more prone to over-fitting.

**Textual inversion.** To reduce the number of unrealistic images, we explored a variant of TEXTUAL INVERSION (Gal et al., 2022). This method learns a pseudoword $n$ representing the class concept combined with a randomly drawn textual description of the generated image style (e.g., "a photo of a $n$", "a rendering of a $n$", etc.; see Gal et al. (2022) for a full list). While this improves the photorealism of the generated samples (Figure 4F), it only slightly improves the augmentation accuracy over the simple fine-tuning of conditioning vectors (FT CONDITIONING) by 0.1 percentage points (Table 1).

**Fine-tuning the denoising.** We now explore fine-tuning the DMs denoising module jointly with the conditioning vector. This idea stems from IMAGIC (Kawar et al., 2022) and results in an augmentation accuracy of $61.0 \pm 0.3$ (examples in Figure 4G), which is on par with TEXTUAL INVERSION.

Finally, we combine the best-performing methods: jointly optimizing a pseudo-word per class (TEXTUAL INVERSION (Gal et al., 2022)) and the DMs denoising module (IMAGIC (Kawar et al., 2022)). We denote this method by PSEUDOWORD+DM. It resulted in an accuracy of $61.2 \pm 0.2$, outperforming all other DM based techniques investigated. The generated images also show improved photorealism over TEXTUAL INVERSION and IMAGIC (Figure 4H).

**Summary.** Personalizing diffusion models improves in matching the domain of the 10% ImageNet images better. For example, ImageNet was collected about two decades ago, and while prompt-based techniques generate instances of the class "desktop computer" similar to modern computers (last row in Figure 4B-D), the fine-tuned models generate images depicting older computers (Figure 4F-H), better matching images from the time when ImageNet was created (Figure 4A; more examples in Appendix Figure 7). Furthermore, personalizing improves upon prompt-based techniques to reduce class-ambiguity (e.g. papillon and desktop computer showed ambiguity in Figure 4B but not in panels E-H), show good sample variety and the best-performing PSEUDOWORD+DM method enhances photorealism and reduces the artistic style of generated images. This model performs best across all generative augmentation techniques investigated, and suggests that we can leverage personalization techniques to combine the DM's knowledge of billions of annotated images with learning the domain distribution of the data we want to augment. At the same time, none of these sophisticated methods outperformed the simpler and computationally cheaper RETRIEVAL baseline.

## 4.5 When does retrieval help?

**Retrieval helps for most classes.** To better understand the failure cases of the augmentation methods, we check for each class in ImageNet if the augmentation method is beneficial. To this end, for each class, we compute the performance improvement of the downstream classifier using the augmentation method compared to only using the original ImageNet samples. Figure 5A shows the distribution of improvements for

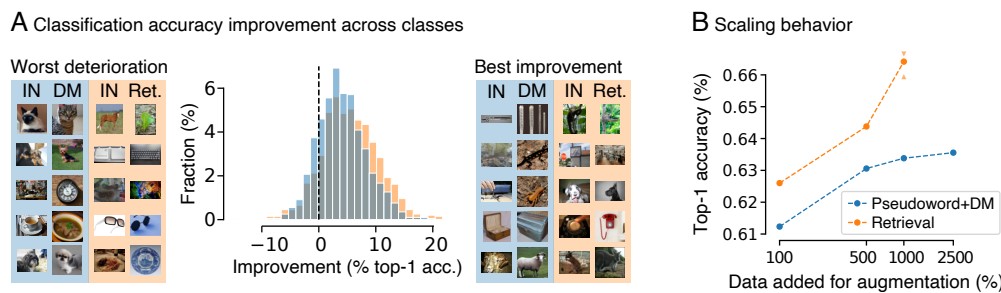

Figure 5: *Retrieval and diffusion model performance across classes and scaling behavior.* **A.** The distribution of classifier improvement for each class is shown for the best-performing DM-based method PSEUDOWORD+DM (blue) and RETRIEVAL (orange). For each class, the improvement is computed by comparing the downstream classification performance using augmentations compared to only using the original 10% ImageNet samples. Images to the left and right show examples of classes with the worst deterioration and greatest improvement. **B.** RETRIEVAL outperforms synthetic images across augmentation ranges. For 1000% augmentation, we could not retrieve sufficient samples for 7 additional classes. Assuming classifier performance of 0% or 100% on these classes leads to best- (▼) and worst-case (▲) estimates.

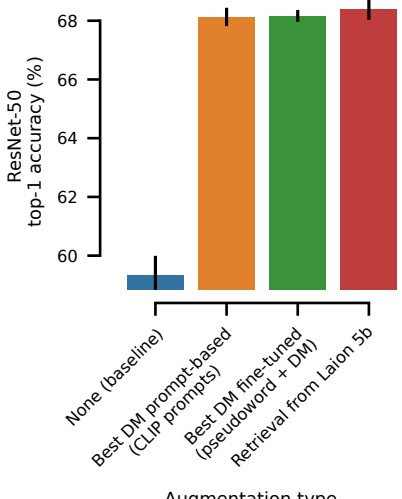

Figure 6: *Results on Caltech256.* Similarly to ImageNet, on Caltech256 diffusion model based approaches did not outperform our proposed simple RETRIEVAL baseline as well. Error bars: standard error of the mean across five runs.

the RETRIEVAL augmentation method and the best DM-based method PSEUDOWORD+DM. Both methods improve the performance for most of the classes, however, there are still many classes where the performance is decreased up to 10%. The distribution of improvements for RETRIEVAL is similarly shaped as for PSEUDOWORD+DM, but significantly pushed to the right. Systematically investigating these failure cases might be a fruitful avenue to further improve generative augmentation.

**Retrieval has better scaling behavior.** As diffusion models can generate arbitrarily many samples, we explored our best-performing diffusion model's (PSEUDOWORD+DM) scaling behavior for higher augmentation ratios. To compare its performance to RETRIEVAL, we retrieved abundant samples and augmented with those images which CLIP embeddings were most similar to our retrieval prompt (Figure 5B). To reach the performance obtained by adding 100% retrieved images would require augmenting with approximately 400% to 500% synthetic images (estimated from Figure 5B), highlighting retrieval's data efficiency. As more data is added augmenting with samples from both methods, the performance gap between augmentation with retrieved and synthetic images increases substantially. Since the performance of classifiers trained on synthetic images saturates, it is unlikely that there will be sufficient data for the current generative methods to close the gap in a like-with-like comparison.

**Augmenting full datasets.** So far, all presented results address the case of a small dataset, simulated by drawing a subsample from ImageNet. Hence, we asked how well the best prompt-conditioned and personalized DM augmentation techniques would perform compared to retrieval on large datasets. On full ImageNet (Figure 1B), we found results consistent with the subsampled ImageNet: RETRIEVAL performed best ($79.0 \pm 0.2$) followed by PSEUDOWORD+DM ($78.8 \pm 0.2$) and CLIP PROMPTS ($78.7 \pm 0.1$). All of our approaches outperformed the ResNet-50 baseline (76.1) and the concurrent study by Azizi et al. (2023) (78.17% ResNet-50 accuracy, see their Table 3, column "Real+Generated").

On full ImageNet, the performance boost of all methods narrowed compared to the data-scarce case: RETRIEVAL improved 2.9% accuracy on full ImageNet vs. 5.4% points on 10% ImageNet; similarly, PSEUDOWORD+DM improved 2.7% on full ImageNet vs. 4% on 10% ImageNet (see Figure 1A vs. 1B). This does not mean that retrieval scales worse than DM based approaches. Rather, we generally expect augmentation to be less effective on large datasets, which would explain the smaller performance gains (Hyams et al., 2019).

Additionally, we verified our results on Caltech256 (Griffin et al., 2007), showing that diffusion models do not outperform our simple RETRIEVAL baseline across datasets (Figure 6; RETRIEVAL: $68.4 \pm 0.4$, PSEUDOWORD+DM: $68.1 \pm 0.3$, CLIP PROMPTS: $68.2 \pm 0.2$, no augmentation: $59.3 \pm 0.7$).

**Retrieval adds negligible ImageNet duplicates.** Cherti et al. (2022) reported that one percent of ImageNet images are contained in Laion 400m, a predecessor of the larger Laion 5b used in our study, raising two questions: (1) did RETRIEVAL perform better than diffusion models because it added real ImageNet images that were not available for training before, and more importantly (2) did test-set images leak into the augmented training set? To address these questions, we leveraged perceptual image hashing (Zauner, 2010) to compare images used for augmentation to ImageNet samples and considered any image-pair as a duplicate candidate if their hashes' Hamming distance was less than ten (Cherti et al., 2022). Next, we refined this approach by manually inspecting all duplicate candidates. For 10% ImageNet, we found 0.009% of the test-set (11 of 126,861 test images) leaked to the RETRIEVAL augmented training data (Appendix Figure 8). If we conservatively assume that these 11 test-images were classified correctly when included in the training-set, but would be misclassified if they were not present in the training-set, we would see a drop of 0.009% accuracy. 0.13% of the retrieved images were either in our validation split or the portion of ImageNet not used in the 10% training subsample. On the original test-set, which we did not use in our study, hashing found 2,377 potential duplicates. We inspected 500 of them and found four actual duplicates. Hence, we estimated 19 duplicates (0.01% of the retrieved data) were used for augmentation. We hypothesize original test-set duplicates might be scarcer since original test-labels were never made public, thus, they might be inaccessible in the text to image nearest neighbor search index. For the best performing diffusion model (PSEUDOWORD+DM) we inspected 1,000 potential duplicates and did not find identical images (examples in Appendix Figure 9). Overall, the minimal overlap of retrieved images and ImageNet cannot account for RETRIEVAL's large performance boost of 1.4% over PSEUDOWORD+DM.

For full ImageNet, we ran RETRIEVAL at a later point, and different images were returned because accessibility of the images at the internet addresses provided by Laion 5b fluctuates (see Section 3.1 for details). Here, we found that 0.002% test images (3 of 126,861 test images) leaked into the augmented data. The only relevant influx of additional ImageNet images to the full ImageNet train-set could emerge from the validation set (we found 2 images, corresponding to 0.0002% of retrieved samples) or from the original test-set, for which we estimated that 187 real duplicates (0.02% of the retrieved data) were added by augmentation (estimated based on the true duplicate rate obtained in the previous paragraph). Again, these minimal numbers of duplicates cannot account for RETRIEVAL's superior performance, adding 0.2% accuracy over PSEUDOWORD+DM. Overall, our results are in line with previous studies reporting only minor impact of duplicates (Radford et al., 2021; Zhai et al., 2022; Li et al., 2023).

## 5   Discussion

Diffusion models have shown their effectiveness in many application areas, and using them for data augmentation is an intriguing research direction. We have evaluated multiple methods to prompt and personalize

diffusion models on their usefulness for data augmentation, showing that none of them beat the simple baseline of retrieving images from the diffusion model's pre-training dataset. Why is the retrieval baseline so hard to beat? One reason might be that retrieval potentially accesses more information, as the pre-training dataset is usually much larger than the weights of the generative models trained on it. Although it has been argued that diffusion models might partially compress the training data (Somepalli et al., 2022; Carlini et al., 2023), it is still unclear if the generative model captured all relevant information. However, diffusion models could possibly improve upon the so-far superior retrieval by generating a large amount of additional data and more diverse and compositionally novel images, for instance by generating out-of-domain samples (e.g., "a photograph of an astronaut riding a horse") (Ramesh et al., 2022; Saharia et al., 2022b; Rombach et al., 2022). Furthermore, diffusion models allow, in principle, for more controlled adaptation than retrieval methods. We showed that personalization methods are a good step in this direction, however, they typically focus only on creating variants of specific given images. To unlock the true potential of diffusion models for data augmentation, new methods that capture the target dataset manifold as a whole, are needed.

**Limitations and future work.** Diffusion models are an active research area, and new methods and applications are published on a daily basis. Hence, our experiments could only capture a subset of possible methods and newly proposed extensions of them (in total, eleven methods), and we evaluated all methods by training a ResNet-50 classifier. However, performance might vary for other classifier choices. Similarly, replacing DM backbones by newer, better versions might improve performance for any augmentation method that is based on them. To simulate a regime of scarce data, in which augmentation generally is most helpful, we used the same 10% random ImageNet subsample throughout our experiments. In principle, selecting another subsample from the same data distribution might slightly alter classifier performance. However, as this sub-split is shared across all augmentation methods, all models (including the baselines) would be systematically affected in the same way. Thus, we do not expect the particular choice of a specific sub-split would relevantly change performance differences between methods or affect our overall results. This notion is in line with our finding of results being consistent between 10% ImageNet and full ImageNet.

It would be interesting to extend our analysis to other specialized data for which collecting new samples might be difficult, for example medical images. Such data might not be available in web-crawled data like Laion 5b on which diffusion models were trained on. Regardless, previous work adapted DMs to medical data, generating privacy-preserving dataset clones. Downstream classifiers trained on these synthetic clones exhibited a performance gap to real data (Packhäuser et al., 2022; Ghalebikesabi et al., 2023). Augmenting by retrieved images might be even more severely impacted, as appropriate images might not be sufficiently available in the web-crawled data. Evaluating and developing generative and retrieval based augmentation methods for such specialized data might be an impactful avenue for future work. Similarly, exploring the investigated methods for out-of-distribution generalization might be promising future work as well (Qiu et al., 2022; Wenzel et al., 2022a). We have shown that simple retrieval is a very strong competitor for data augmentation. Further improvements could be introduced by diversifying the retrieval set (Wenzel et al., 2020; Yue & Guestrin, 2011), retrieving images using linear combinations of inputs (Zietlow et al., 2022), or generating more diverse retrieval search prompts leveraging off-the-shelf word-to-sentence models and data-filtering that were beneficial in a CLIP fine-tuning setting (He et al., 2023).

**Conclusion.** We showed that a simple, computationally efficient retrieval baseline outperforms a wide range of diffusion-model-based augmentations. However, given the fast rate of progress in this field it is not possible to definitively say that retrieval can not be beaten, and we believe that diffusion models have the potential to improve over this baseline. Nonetheless, the strength of retrieval's performance makes it clear that future works using diffusion models for augmentation should also compare against this baseline. We hope that our paper provides ground for researchers to benchmark generative augmentation methods and assess their benefit by comparing them with retrieval baselines. In the longer term, we hope that new methods can be developed that combine retrieval and generation, leading to greater improvements in the diversity and quality of augmented images.

## 6 Broader impact

As our work uses Laion 5b (Schuhmann et al., 2022) and stable diffusion (Rombach et al., 2022), many of the ethical and societal problems of these works propagate to our study. For example, explicit images of rape, pornography, stereotypes, racist and ethnic biases, and other toxic content have been found in Laion 400m (Birhane et al., 2021) and it is very likely that they are contained in its larger successor Laion 5b. As stable diffusion was trained on similar data, it might produce similarly problematic content (Carlini et al., 2023). We filtered images by the automatic safety-checkers provided by Laion 5b and stable diffusion and trained classifiers on them. As such, offensive images, even those missed by the safety checkers, will not be presented to users of the classifier. However, systematic biases present in the data are still likely to be present in the final classifier decisions. This is less of a concern for the sanitized labels used in the ILSVRC2012 ImageNet challenge which we evaluated on (the only person categories used are "scuba diver", "bridegroom", and "baseball player"; Yang et al. (2020)) but for potential other tasks not studied in our work, such as labeling people as "hard-worker" vs. "lazy", racial and gender biases should be expected. These concerns are not unique to the use of Laion 5b, and have been shown to be present in a wide range of web-sourced datasets. In particular, the person subtree that used to be contained in the larger ImageNet dataset (this is a superset of the data used by the ImageNet Large Scale Visual Recognition Challenge (ILSVRC2012) that we considered in our study) has been shown to contain offensive labels and biases Crawford & Paglen (2019); Yang et al. (2020).

## Acknowledgements

The authors would like to thank Varad Gunjal and Vishaal Udandarao. MFB thanks the International Max Planck Research School for Intelligent Systems (IMPRS-IS).

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
