# OpenReview forum: "Image retrieval outperforms diffusion models on data augmentation"
_TMLR — Accepted by TMLR_

### Review · Reviewer_Kp2i · 2023-09-04

**Summary Of Contributions:**

Summary: The paper presents a study of different diffusion model-based methods to identify effective strategies for data augmentation. The authors evaluate all the baselines in the context of classification accuracy on the ImageNet benchmark. Surprisingly, the authors find that a simple image-retrieval-based baseline outperforms more sophisticated diffusion model-based augmentation strategies on classification performance.

**Audience:**

Yes

**Claims And Evidence:**

Yes

**Requested Changes:**

Requested Changes/Clarifications:

1) Is it possible for the LAION-5B dataset to overlap with ImageNet? If yes, one potential concern is that the simple image-retrieval-based baseline might be using data from the ImageNet training set, in which case I would expect the performance of this baseline to be better than a dataset augmented with synthetic samples generated from a diffusion model. It would be nice if the authors could elaborate on this aspect in the paper when mentioning the retrieval-based baseline in Section 3.

2) I think the difference between the retrieval baseline and the diffusion model baselines for both full ImageNet and Caltech datasets (as reported in Section 4.5) is insignificant (or marginal at best). In that sense, does this mean that the retrieval-based approach scales worse (in terms of dataset size) than the diffusion model-based augmentation methods and might be primarily valuable for augmentation in small datasets (which is also a valuable contribution)? If yes, I think it would be better to combine the sub-sections: `Retrieval has better-scaling behavior` and `Retrieval works on full data sets` into one sub-section describing the overall scaling behavior. Moreover, this should also be reflected in the claims that the authors presently make which currently indicate that the retrieval baseline outperforms diffusion model-based augmentation by a large margin when that might not be the case (as per the results in Section 4.5 on the full ImageNet dataset).

3) Can the authors include some additional experiments on the 10% ImageNet subset but using a more recent classifier architecture like ViT (smaller variants) in the Appendix (just to rule out the possibility of results being specific to a classifier architecture)

4) Are the diffusion model variants 5 (FT Conditioning) and 6 (FT Cluster Conditioning) proposed in this paper, as I do not see any citations for these baselines?

**Strengths And Weaknesses:**

Strengths:

1) Given the advances in synthetic data generation using diffusion models, this is a timely contribution to identifying effective strategies for data augmentation using large-scale pre-trained diffusion models like StableDiffusion.

2) I found the overall experimental setup clear. The discussion in Section 4 is overall interesting (especially the discussion of different baselines for generating synthetic data).

Weaknesses:
See Requested Changes below

---

> ### Author Response · Authors · 2023-10-19
>
> > Is it possible for the LAION-5B dataset to overlap with ImageNet? If yes, one potential concern is that the simple image-retrieval-based baseline might be using data from the ImageNet training set, in which case I would expect the performance of this baseline to be better than a dataset augmented with synthetic samples generated from a diffusion model. It would be nice if the authors could elaborate on this aspect in the paper when mentioning the retrieval-based baseline in Section 3.
>
> Thank you very much for raising this important concern. The Laion authors and colleagues (Cherti et al., 2022) found that 1.02% of ImageNet images were contained in Laion 400m, the predecessor of the larger Laion 5b used in our study, further underpinning your concern. Specifically, this raises two questions: (1) did retrieval perform better than diffusion models because it added real ImageNet images to the augmented dataset that were not available for training before, and more importantly (2) did test set images leak into the augmented training set? Hence, we now have repeated the analysis of Cherti et al. (2022) to identify duplicates between Laion 5b and ImageNet by perceptual image hashing (pHash; Zauner, 2010). We additionally refined their approach by manual inspection of automatically pHash flagged duplicate candidates. We found that (1) overlap of the retrieved images to ImageNet and (2) test-image leaking was negligible, a result overall in line with previous studies reporting only minor impact of duplicates (Radford et al., 2021; Zhai et al., 2022; Li et al., 2023). We now exactly describe this analysis and its results in paragraph “Retrieval adds negligible ImageNet duplicates.” in Section 4.5 and depict example duplicate candidates in new supplementary Figures 8 and 9. Thank you very much again for pointing this out, as the now included analysis you suggested makes our paper stronger.
>
> Zauner, C. Implementation and benchmarking of perceptual image hash functions. (2010).
>
> Cherti, M. et al. Reproducible scaling laws for contrastive language-image learning. arXiv.org https://arxiv.org/abs/2212.07143v1 (2022).
>
> Alec Radford, Jong Wook Kim, Chris Hallacy, Aditya Ramesh, Gabriel Goh, Sandhini Agarwal, Girish Sastry, Amanda Askell, Pamela Mishkin, Jack Clark, et al. Learning transferable visual models from natural language supervision. In International Conference on Machine Learning, pages 8748–8763. PMLR, 2021
>
> Xiaohua Zhai, Xiao Wang, Basil Mustafa, Andreas Steiner, Daniel Keysers, Alexander Kolesnikov, and Lucas Beyer. LiT: Zero-shot transfer with locked-image text tuning. In Proceedings of the IEEE/CVF Conference on Computer Vision and Pattern Recognition, pages 18123–18133, 2022
>
> Li, A. C., Brown, E. L., Efros, A. A. & Pathak, D. Internet Explorer: Targeted Representation Learning on the Open Web. in Proceedings of the 40th International Conference on Machine Learning 19385–19406 (PMLR, 2023).

---

> > ### Comment · Reviewer_JdHi · 2023-10-20
> >
> > > We found that (1) overlap of the retrieved images to ImageNet and (2) test-image leaking was negligible, a result overall in line with previous studies reporting only minor impact of duplicates (Radford et al., 2021; Zhai et al., 2022; Li et al., 2023). We now exactly describe this analysis and its results in paragraph “Retrieval adds negligible ImageNet duplicates.” in Section 4.5 and depict example duplicate candidates in new supplementary Figures 8 and 9. Thank you very much again for pointing this out, as the now included analysis you suggested makes our paper stronger.
> >
> > Thank you very much for the reviewer to raise this critical point.
> >
> > So, the proposed model has been trained by the ImageNet test images? Then what are we talking about? What is negligible? Could you provide more statistics on the number of ImageNet images you used during training or at test time? Experiments should be repeated after removing those images from training and testing, no?

---

> > > ### Author Response · Authors · 2023-10-20
> > >
> > > > Thank you very much for the reviewer to raise this critical point.
> > > >
> > > > So, the proposed model has been trained by the ImageNet test images? Then what are we talking about? What is negligible? Could you provide more statistics on the number of ImageNet images you used during training or at test time? Experiments should be repeated after removing those images from training and testing, no?
> > >
> > > 10% ImageNet corresponds to 126,861 training images, which we augmented by adding the same number of retrieved images. The test-set always contained 126,861 images disjoint with training and validation sets. Here, 11 test images leaked into the training set (paragraph “Retrieval adds negligible ImageNet duplicates.” in Section 4.5).  If we conservatively assume that these 11 test-images were classified correctly when included in the training-set, but would be misclassified if they were not present in the training-set, we would see a drop of 0.009% in accuracy (11 of 126,861 misclassified). Similarly, for the experiments on full ImageNet, 3 test images leaked. Under the same conservative assumptions this would correspond to 0.002% accuracy drop (3 of 126,861 misclassified). As retrieval added accuracy boosts 3 orders of magnitude higher (+5.4% over the baseline on 10% ImageNet and +2.9% on 100% ImageNet; see Figure 1), test-image leaking is negligible, and due to the high performance boosts the outcomes of our experiments would not change by removing those images from training.
> > >
> > > We now have revised paragraphs “Dataset.” in Section 3 and “Retrieval adds negligible ImageNet duplicates.” in Section 4.5 to state this more explicitly and we thank you very much for raising this clarifying question.

---

> > > > ### Comment · Reviewer_JdHi · 2023-10-20
> > > >
> > > > Thanks for your reply.
> > > > 11 test images leaked into the training set.
> > > > Sorry but I'm not sure whether the math for accuracies makes sense. The network was able to learn from those 11 test images during training. For instance if there was domain gap, the network could cover it.
> > > > Also, I don't understand why there were less images leaked when using the full ImageNet, it should be only more, no?

---

> > > > > ### Author Response · Authors · 2023-10-23
> > > > >
> > > > > Thank you for following up on this.
> > > > >
> > > > > We retrieved data to train the classifiers on the 10% ImageNet subsplit first, and at a later point the classifiers on full ImageNet. Laion 5b does not provide the images itself, but rather internet addresses to download the images from. As some of these links expired, different samples were used on full ImageNet, hence resulting in slightly different duplicate numbers. (2nd paragraph at "Retrieval adds negligible ImageNet duplicates.” in Section 4.5 and paragraph “Retrieval is computationally efficient.” in Section 3.1.)
> > > > >
> > > > > All methods using Laion or diffusion models had access to the same pool of large data and small amounts of intersection between datasets is now common with web scale data. As such, performance should be seen as being representative in this common scenario where large datasets are not thoroughly curated and can slightly overlap. There are many previously accepted papers that operate in the same setting, and many of them found results were not relevantly impacted by that small overlap (just to provide a few examples: CLIP: Radford et al, 2021; papers [A], [B], and [C] you mentioned before; Cherti et al., 2022; Zhai et al., 2022; Li et al., 2023). Overall, in this field it is common to acknowledge that web scale data leads to minor overlaps, and the large number of published papers reflects that the community finds these results worthwhile to be published.
> > > > >
> > > > > We hope our replies have further clarified your questions.
> > > > >
> > > > > [A] He et al. "Is synthetic data from generative models ready for image recognition?", ICLR 2023
> > > > >
> > > > > [B] Azizi et al. "Synthetic Data from Diffusion Models Improves ImageNet Classification", CVPR 2023
> > > > >
> > > > > [C] Sariyildiz et al. "Fake it till you make it: Learning transferable representations from synthetic ImageNet clones", CVPR 2023
> > > > >
> > > > > Cherti M, Beaumont R, Wightman R, Wortsman M, Ilharco G, Gordon C, Schuhmann C, Schmidt L, Jitsev J. Reproducible scaling laws for contrastive language-image learning. In Proceedings of the IEEE/CVF Conference on Computer Vision and Pattern Recognition 2023 (pp. 2818-2829).
> > > > >
> > > > > Alec Radford, Jong Wook Kim, Chris Hallacy, Aditya Ramesh, Gabriel Goh, Sandhini Agarwal, Girish Sastry, Amanda Askell, Pamela Mishkin, Jack Clark, et al. Learning transferable visual models from natural language supervision. In International Conference on Machine Learning, pages 8748–8763. PMLR, 2021
> > > > >
> > > > > Xiaohua Zhai, Xiao Wang, Basil Mustafa, Andreas Steiner, Daniel Keysers, Alexander Kolesnikov, and Lucas Beyer. LiT: Zero-shot transfer with locked-image text tuning. In Proceedings of the IEEE/CVF Conference on Computer Vision and Pattern Recognition, pages 18123–18133, 2022
> > > > >
> > > > > Li, A. C., Brown, E. L., Efros, A. A. & Pathak, D. Internet Explorer: Targeted Representation Learning on the Open Web. in Proceedings of the 40th International Conference on Machine Learning 19385–19406 (PMLR, 2023).

---

> ### Author Response · Authors · 2023-10-19
>
> > I think the difference between the retrieval baseline and the diffusion model baselines for both full ImageNet and Caltech datasets (as reported in Section 4.5) is insignificant (or marginal at best). In that sense, does this mean that the retrieval-based approach scales worse (in terms of dataset size) than the diffusion model-based augmentation methods and might be primarily valuable for augmentation in small datasets (which is also a valuable contribution)? If yes, I think it would be better to combine the sub-sections: `Retrieval has better-scaling behavior` and `Retrieval works on full data sets` into one sub-section describing the overall scaling behavior. Moreover, this should also be reflected in the claims that the authors presently make which currently indicate that the retrieval baseline outperforms diffusion model-based augmentation by a large margin when that might not be the case (as per the results in Section 4.5 on the full ImageNet dataset).
>
> We agree that the performance difference between retrieval and the diffusion models (DM) are minor on the full datasets (Figures 1B and 6). We have added text to the introduction and elsewhere emphasizing this. Such behavior is expected. Just as adding more data becomes less effective in high-data regimes (Hyams et al., 2019), the choice of augmentation method also becomes less important. Even in such situations, we believe the simplicity and computation efficiency of retrieval (simply look up a CLIP embedding in a pre-existing index, and download the according image) make it a compelling alternative to diffusion models, which also was not yet outperformed by diffusion-model-based approaches. Investigating scaling from a different perspective – instead of increasing original dataset size, we now consider increasing the number of augmented samples – shows that performance scales much better for retrieval compared to diffusion models (Figure 5B).
>
> We now have added an extensive discussion of dataset sizes to paragraph “Augmenting full datasets” in Section 4.5 and improved the description of scaling behavior (ie. scaling number of original training samples vs. number of samples added by augmentation). Additionally, to present our results more precisely, we now have added results on full ImageNet to new Figure 1B and Caltech256 in new Figure 6, and in the Introduction we have revised the penultimate paragraph and added a new paragraph. Likewise, we generally improved on clearly relating our results to the data used throughout the paper (first paragraph in Results, paragraph “Limitations and future work” in the Discussion, and elsewhere). Thank you for your comment, which helped us to better present the context on which our claims are based on.
>
> Gal Hyams, Dan Malowany, Ariel Biller, and Gregory Axler. Quantifying diminishing returns of annotated data: How much data do you really need?, 2019. https://clear.ml/blog/quantifying-diminishing-returns . Accessed: 19 Oct 23.

---

> ### Author Response · Authors · 2023-10-19
>
> > Can the authors include some additional experiments on the 10% ImageNet subset but using a more recent classifier architecture like ViT (smaller variants) in the Appendix (just to rule out the possibility of results being specific to a classifier architecture).
>
> Thank you very much for suggesting these additional experiments. Unfortunately, we now have very limited access to compute as we used most of our compute budget on the large number of compute intense diffusion model augmentation strategies. Thus, we cannot run ablation studies for additional classifier choices, similar to other studies massively generating images from diffusion models for various use cases (Luzi et al., 2022; Sariyildiz et al., 2022; Roy et al., 2022; Packhäuser et al., 2022). We sincerely apologize for being unable to run these additional experiments. We now have carefully revised our paper to clearly state that any claims were obtained for ResNet-50 only and not ablated over classifier architectures (new axis labels in Figures 1 and 6, caption in Figure 1, penultimate paragraph in Introduction, paragraph “Data augmentation and classifier training.” in Section 3, first paragraph in the Results (Section 4), caption of Table 1). In addition, we now explicitly mention that performance numbers might differ across various classifier architectures in paragraph “Limitations and future work.” in the Discussion section.
>
> Luzi, L., Siahkoohi, A., Mayer, P. M., Casco-Rodriguez, J. & Baraniuk, R. Boomerang: Local sampling on image manifolds using diffusion models. Preprint at http://arxiv.org/abs/2210.12100 (2022).
>
> Sariyildiz, M. B., Alahari, K., Larlus, D. & Kalantidis, Y. Fake it till you make it: Learning(s) from a synthetic ImageNet clone. Preprint at http://arxiv.org/abs/2212.08420 (2022).
>
> Roy, A., Shah, A., Shah, K., Roy, A. & Chellappa, R. DiffAlign : Few-shot learning using diffusion based synthesis and alignment. Preprint at http://arxiv.org/abs/2212.05404 (2022).
>
> Packhäuser, K., Folle, L., Thamm, F. & Maier, A. Generation of Anonymous Chest Radiographs Using Latent Diffusion Models for Training Thoracic Abnormality Classification Systems. Preprint at http://arxiv.org/abs/2211.01323 (2022).
>
>
>
> > Are the diffusion model variants 5 (FT Conditioning) and 6 (FT Cluster Conditioning) proposed in this paper, as I do not see any citations for these baselines?
>
> You are right, the models you mentioned are proposed in our paper. We now have revised Table 1 and paragraphs “Prompt conditioning.”, “Fine-tuning the diffusion model.”, and “Laion nearest neighbor retrieval.” in Section 3.1 to clearly state which methods we propose. Thank you very much for bringing this up, as we believe this now made our paper stronger.

---

### Review · Reviewer_ixUD · 2023-09-06

**Summary Of Contributions:**

The paper benchmarks diffusion models for data augmentation. In particular, it compares the performance achieved using different data augmentation strategies, including different prompting strategies and nearest neighbor retrieval. The paper finds that a suggested baseline using nearest neighbor retrieval on CLIP embeddings outperforms other methods.

**Audience:**

Yes

**Broader Impact Concerns:**

I do not see any need to have a broader impact statement.

**Claims And Evidence:**

Yes

**Requested Changes:**

- Minor: I would recommend to merge the sections 3.1 and 3.2 and explain the details of each method right after it was introduced. For example, currently the "Laion nearest neighbor retrieval" is lacking detail and is hard to understand in section 3.1.

- Please clarify my questions in the "Weaknesses" paragraph

**Strengths And Weaknesses:**

## Strengths:
- The paper is well written and easy to follow.
- The experiments show relevant baselines (e.g. 20% of ImageNet)
- The experiment design is simple but I liked the straightforward approach of testing different augmentation strategies.
- The paper provides an interesting baseline for future evaluation of data augmentation methods.

## Weaknesses:

- The paper shows that the nearest neighbor retrieval baseline outperforms other diffusion methods.
However, I think the usage of CLIP embeddings can be a major limitation hidden in the design of the experiments. CLIP was trained on pairs of images and their textual description crawled from the internet. Obviously, these embeddings are mostly trained on natural images. I would therefore question that the approach would work on more specialized datasets, such as medical images. However,
specialized dataset are exactly the ones where training would benefit the most from data augmentation.
While medical images are shortly mentioned in the limitation sections, this issue should be discussed in more detail in the paper.


- The description of the used data is unclear. "We resample the additional data into 5 sets containing the same number of samples per class as our target dataset and train a ResNet-50 classifier (He et al., 2016) on each to derive variance estimates." Is the training on an generated clone of the original dataset or on the original dataset with additional samples? Please clarify this! The term "augmentation" would suggest that the original dataset is used. However, as I understand it, this is not the case?


## Questions

- In section 3, paragraph "Data augmentation" it is stated that for ImageNet 390 samples per class are sampled. However, in section 3.2 paragraph "nearest neighbor retrieval" it is stated that
130 images per class for ImageNet were retrieved. Please clarify these mismatching numbers.

- How long did the training take on a the NVIDIA T4 and V100 GPUs? (Useful to estimate the cost for reproducing the experiments)

---

> ### Author Response · Authors · 2023-10-19
>
> > The paper shows that the nearest neighbor retrieval baseline outperforms other diffusion methods. However, I think the usage of CLIP embeddings can be a major limitation hidden in the design of the experiments. CLIP was trained on pairs of images and their textual description crawled from the internet. Obviously, these embeddings are mostly trained on natural images. I would therefore question that the approach would work on more specialized datasets, such as medical images. However, specialized datasets are exactly the ones where training would benefit the most from data augmentation. While medical images are shortly mentioned in the limitation sections, this issue should be discussed in more detail in the paper.
>
> Thank you for bringing to our attention that the issue of augmenting specialized data, such as medical images, should be discussed in more detail. Both, diffusion models and retrieval, rely on CLIP embeddings (diffusion models use them for guided generation). As such, both are limited by CLIP being trained on natural images. However, diffusion models can be adapted to medical data to generate privacy preserving clones of the real data set, although leading to a performance gap (Packhäuser et al., 2022; Ghalebikesabi et al., 2023). While CLIP when used for retrieval could be fine-tuned on medical data as well, it might still be difficult to retrieve such images because they might not be available in web-crawled data sets like Laion 5b. We now discuss this more extensively in addition to citing related papers in paragraph “Limitations and future work.” in the Discussion section of the paper.
>
> Packhäuser, K., Folle, L., Thamm, F. & Maier, A. Generation of Anonymous Chest Radiographs Using Latent Diffusion Models for Training Thoracic Abnormality Classification Systems. Preprint at http://arxiv.org/abs/2211.01323 (2022).
>
> Ghalebikesabi, S. et al. Differentially Private Diffusion Models Generate Useful Synthetic Images. Preprint at http://arxiv.org/abs/2302.13861 (2023).
>
>
> > The description of the used data is unclear. "We resample the additional data into 5 sets containing the same number of samples per class as our target dataset and train a ResNet-50 classifier (He et al., 2016) on each to derive variance estimates." Is the training on a generated clone of the original dataset or on the original dataset with additional samples? Please clarify this! The term "augmentation" would suggest that the original dataset is used. However, as I understand it, this is not the case?
>
> Thank you for pointing out this issue. We added each of the five additional datasets to the original training data for augmentation, resulting in five augmented datasets that were used for classifier training. We now have described the data clearer in the paragraph you mentioned (paragraph “Data augmentation and classifier training.” in section “Experimental protocol”).
>
>
> > In section 3, paragraph "Data augmentation" it is stated that for ImageNet 390 samples per class are sampled. However, in section 3.2 paragraph "nearest neighbor retrieval" it is stated that 130 images per class for ImageNet were retrieved. Please clarify these mismatching numbers.
>
> Another great suggestion to improve the clarity of our paper! Generating images with stable diffusion is stochastic due to the random initialization of the denoising process, leading to a variety of images even though the conditioning is kept fixed. To account for that variability, we obtained a variance estimate for the downstream classifier performance by generating 390 samples (three-fold the size of the target data). We then randomly subsampled five augmented data sets for classifier training, yielding the desired performance variance estimates. On the other hand, retrieving images from Laion 5b is deterministic: prompting the Laion 5b nearest neighbor search index leads to those 130 images that represent the prompt text best in terms of their CLIP similarity score. Here, the data itself does not require a variance estimate, and varying classifier performance is solely caused by random initialization of the five ResNet-50 training runs. We now have improved the description of paragraph “Data augmentation and classifier training.” in Section 3 and paragraph “Laion nearest neighbor retrieval.” in new Subsection 3.1.

---

> ### Author Response · Authors · 2023-10-19
>
> > How long did the training take on the NVIDIA T4 and V100 GPUs? (Useful to estimate the cost for reproducing the experiments)
>
> Thank you for raising this question. Checking log files and file time-stamps we now have estimated training times for experiments with 10% ImageNet. For fine-tuning the diffusion models, training times depended on the specific method, and varied from 244 GPU hours to 720 GPU h (trained on 8 Nvidia V100 GPUs using distributed data parallel training). Generating 390 images per ImageNet class required 867 GPU h on a Nvidia T4 GPU. Training the ResNet-50 classifier was early stopped on the validation accuracy, hence, training times varied and were typically between 128 and 160 GPU h (trained on 8 Nvidia T4 GPUs using distributed data parallel training). We now have revised paragraph  “Laion nearest neighbor retrieval.” and have added a new paragraph “Retrieval is computationally most efficient.” providing detailed compute requirements (both in new Section 3.1).
>
>
> > Minor: I would recommend to merge the sections 3.1 and 3.2 and explain the details of each method right after it was introduced. For example, currently the "Laion nearest neighbor retrieval" is lacking detail and is hard to understand in section 3.1.
>
> We now have merged Sections 3.1 and 3.2 and believe this substantially improved readability of the “Augmentation methods” Section, and we thank you for this suggestion.
>
>
> > Please clarify my questions in the "Weaknesses" paragraph
>
> We replied to each of your questions above and want to express our appreciation for posing them, as they helped us to substantially improve the quality of our paper.

---

### Review · Reviewer_JdHi · 2023-09-12

**Summary Of Contributions:**

This paper explores the efficacy of diffusion models in generating training data for downstream classification tasks on 10% of ImageNet and Caltech256.
A number of different techniques for generating data by diffusion models is compared.
These methods include simple prompting of frozen Stable Diffusion as well as fine-tuning it (i.e., personalizing it) for downstream tasks.
Moreover, a retrieval method is introduced where instead of training a classifier for the downstream task using synthetic data, a set of images retrieved from the training set of Stable Diffusion (LAION-5B), to be used for training the classifier.
Experiments reveal that while fine-tuning Stable Diffusion is better than keeping it frozen, simpler retrieval methods using the model's pre-training data are more effective.

**Audience:**

Yes

**Broader Impact Concerns:**

It is significantly important to openly discuss and acknowledge potential data- and model-related biases in scenarios involving generating synthetic data to tackle specific tasks. Although the proposed method does not directly synthetise images, it utilizes uncurated web-data, which raises similar risks. I suggest the authors to look at recent works in this field like [C], to develop a perspective, and add a Broader Impact Statement accordingly.

[C] Sariyildiz et al. "Fake it till you make it: Learning transferable representations from synthetic ImageNet clones", CVPR 2023

**Claims And Evidence:**

No

**Requested Changes:**

Addressing the weaknesses I listed above would strengthen the work in my view.
However, some of the changes (i.e., revising the claims, or extending the experiments in accordence with the issues I mentioned) might require work that is beyond the two-weeks discussion period or might require evaluating the submission from scratch again.

**Strengths And Weaknesses:**

### Strengths

- Tackles an actively studied problem in generative AI: Using synthetic data for training downstream models. Growing interest in this field.
- A through comparison of 10 methods is presented for generating synthetic data for downstream classification tasks on 10% ImageNet and Caltech256.
- An alternative method is proposed, which is based on retrieving training data for training the downstream classifiers.
- The propsed method achieves superior performance compared to the baselines.

### Weaknesses

- The efficacy of generating data directly for downstream classification tasks has been nicely studied by [A, B].
These papers are cited in the related work section in a very shallow way without positioning the current submission with respect to those works, moreover, they are not considered in the model taxonomy and experiments.
I would like to the authors to provide an in-depth discussion on the comparison against these works, and a comparison against these works in the experiments if possible.

- As I mentioned below, [B] already claims that it is possible to fine-tune a diffusion model to provide synthetic data with improved utility.
I wonder if all the "personalization" techniques evaluated in the experiments are trained under sufficient conditions.
This is especially important because in Figure-4, images generated by different fine-tuned diffusion models look similar.
ImageNet is collected almost two decades ago. Perhaps the domain gap between now and then can be visible for images of certain classes.
It would be nice to show how the output of pre-trained vs. fine-tuned Stable Diffusion change in that regard.

- Following my previous point, I believe that the bold claims made in this paper should be revised clearly mentioning the type and quantity of data used in the experiments.
Currently, the experiments are performed on low-data regime (i.e., 10% of ImageNet and Caltech256) using the same number of additional generated or retrieved images.
Why low-data regime? Why 10%? Would the outcomes still be consistent with more data? How about the variance of the results (i.e., variance due to picking the particular 10% of ImageNet).
How much generated data would be needed to reach the performance of the proposed retrieval scenario?

- The proposed retrieval method requires the whole training set of Stable Diffusion.
I would expect this process to be computally challenging.
So it would be nice to compare compute requirements for all the groups of methods.

[A] He et al. "Is synthetic data from generative models ready for image recognition?", ICLR 2023
Shows that with proper use of prompting, zero-shot and few-shot performance on downstream tasks can be improved using synthetic data from GLIDE.
It is reasonable to expect that the outcomes from this work would translate here, where Stable Diffusion (a better text-to-image model) is used.

[B] Azizi et al. "Synthetic Data from Diffusion Models Improves ImageNet Classification", CVPR 2023
Demonstrates that, if fine-tuned properly, diffusion models can be used to generate data for downstream classification tasks (ImageNet).
Combining such synthetic data with real ones improve performance of the classifier trained solely using real data.

### Some minor points

Overall, I find the quality of text not so great.
Please see below
- The conceptual comparison between GANs and diffusion models in introduction is misleading.
GANs can condition on text too.
There the comparison is mainly between image generative vs text-to-image generative models, and the text should be revised accordingly.
- The first sentence of Section-3 does not read well.
- Figure-1 should be improved by mentioning what is measured in the figure.
- Taxonomy of methods in Figure-2B is not clear, and rather confusing. I don't have a constructive feedback for this, sorry.
- It would be much better to place Figure-3 and Figure-4 before the experiments.

---

> ### Author Response · Authors · 2023-10-19
>
> > The efficacy of generating data directly for downstream classification tasks has been nicely studied by [A, B]. These papers are cited in the related work section in a very shallow way without positioning the current submission with respect to those works, moreover, they are not considered in the model taxonomy and experiments. I would like the authors to provide an in-depth discussion on the comparison against these works, and a comparison against these works in the experiments if possible.
>
>
> Thank you for suggesting to discuss the related papers [A] (He et al.; 2023) and [B] (Azizi et al.; 2023) you mentioned in more detail. The concurrent paper [B] (preprint surfaced within the same week as ours) in fact is very similar to ours. In this study, in one experiment the authors followed the same evaluation protocol as we did: they fine-tuned the Imagen diffusion model (DM) with frozen class name based prompt-conditioning to the target ImageNet data distribution, then they generated images, augmented full ImageNet, and trained a ResNet-50 downstream classifier on it. This allows a direct comparison to our work: [B]’s classifier performed at 78.17% top-1 accuracy (their Table 3, column “Real + Generated”, first row (ResNet-50)). Excitingly, all of our augmentation techniques (new Figure 1B and revised paragraph “Augmenting full datasets.” in Results section) outperformed their method on full ImageNet: our best-performing fine-tuned DM (Pseudoword+DM) outperforms theirs with additional +0.63% accuracy, being similar to their model in tuning the DM and differs to theirs in additionally tuning a pseudoword conditioning whereas [B] used frozen class name conditioning. Interestingly, our CLIP prompts strategy not requiring a fine-tuned DM outperforms [B]’s fine-tuned DM by +0.53%, and retrieval outperforms [B] with the highest performance boost (+0.83%).
>
> [A] is related in leveraging DM generated images to improve classification, however, their evaluation is different: instead of training a classifier from scratch on augmented data (as in [B] and our paper), they use pre-trained CLIP as an already performing base-classifier and improve its performance by fine-tuning the classifier to better match the desired target distribution (e.g. ImageNet). To do so, they train on images generated by a pre-trained, frozen DM, whereas we leverage a variety of DM strategies , including fine-tuning DMs, and retrieval. Overall, their classification setup in fine-tuning CLIP is very different from ours, since we augment data and train classifiers from scratch. While this renders a direct and fair comparison of performance numbers impossible, we now have added an in-depth comparison to their approach to the related work section.
>
> Based on your comment, we extensively revised our paper at several places. We now have added a detailed discussion of [A, B] in the related work section and compare performance to [B] in new Figure 1B and the Results paragraph "Augmenting full datasets.” in Section 4.5. When providing the reference for [A] in your review further below, you mentioned that replacing GLIDE in [A] by a more recent diffusion model backbone (e.g. stable diffusion) might improve their performance – a great idea which we like a lot and which we now have added to paragraph “Limitations and future work” in the Discussion section. Thank you again for your great suggestion, which made our paper stronger.

---

> ### Author Response · Authors · 2023-10-19
>
> > As I mentioned below, [B] already claims that it is possible to fine-tune a diffusion model to provide synthetic data with improved utility. I wonder if all the "personalization" techniques evaluated in the experiments are trained under sufficient conditions. This is especially important because in Figure-4, images generated by different fine-tuned diffusion models look similar. ImageNet was collected almost two decades ago. Perhaps the domain gap between now and then can be visible for images of certain classes. It would be nice to show how the output of pre-trained vs. fine-tuned Stable Diffusion changes in that regard.
>
> Thank you for raising this concern. Fine-tuning our DMs outperformed the prompt-based, frozen DM (Figure 1 and Table 1), and following your previous suggestion, we compared our best performing fine-tuned diffusion model (Pseudoword+DM) to the paper [B] you mentioned, finding that our method outperforms [B] as well. We believe the outperformance of [B], which was previously not discussed in our paper, highlights sound training conditions.
>
> Following your suggestion, we now have added new Figure 7 to the supplement of our paper showcasing that fine-tuning the diffusion model helped in closing the domain gap between ImageNet and the pre-trained stable diffusion model. A similar behavior is depicted in the last row of Figure 4, where ImageNet samples of the class “Desktop computer” show old devices (panel A). Prompting the pre-trained diffusion model (panels B-D) depicts computers that are newer, highlighting the domain gap you mentioned. Interestingly, all fine-tuned DMs resolved this issue, yielding images of older computers similar to images found in ImageNet (panels E-H). This finding was previously not explicitly described in our paper, and we now have added an extensive discussion to highlight this finding in the main text (paragraph “Summary” in Section 4.4), and also refer to the newly added appendix Figure 7 you suggested. Thank you very much for these suggestions which make our paper stronger.
>
>
> > Following my previous point, I believe that the bold claims made in this paper should be revised clearly mentioning the type and quantity of data used in the experiments. Currently, the experiments are performed on a low-data regime (i.e., 10% of ImageNet and Caltech256) using the same number of additional generated or retrieved images. Why a low-data regime? Why 10%? Would the outcomes still be consistent with more data? How about the variance of the results (i.e., variance due to picking the particular 10% of ImageNet). How much generated data would be needed to reach the performance of the proposed retrieval scenario?
>
>
> Thank you for posing these questions. We reply to each of them in turn below, and want to express our gratitude as each of them helped us to significantly improve motivating experiment design and the clarity of how we present the results in our paper.
>
>
> > Why a low data regime?
>
> Augmentation helps the most in scenarios in which data is limited, as is often the case in many real-world scenarios, whereas adding more data becomes less effective in high-data regimes (Hyams et al., 2019). This is reflected by retrieval boosting performance by 5.4% top1 accuracy when augmenting 10% ImageNet (Table 1), whereas retrieval adds 2.9% on full ImageNet (adding the same number of retrieved as available training samples in both cases). Additionally, investigating a large set of augmentation methods (in total eleven) on full ImageNet would be computationally more expensive.  We now revised the paper clearly motivating why we choose a subsample (penultimate paragraph in Introduction, paragraph “Dataset.” in section Experimental protocol). Thank you for pointing out that this was missing before.
>
> Gal Hyams, Dan Malowany, Ariel Biller, and Gregory Axler. Quantifying diminishing returns of annotated data: How much data do you really need?, 2019. https://clear.ml/blog/quantifying-diminishing-returns . Accessed: 19 Oct 23.

---

> ### Author Response · Authors · 2023-10-19
>
> > Variance due to picking the particular 10% of ImageNet
>
> The exact performance numbers might slightly change by picking a different 10% sample of ImageNet, for instance, because the subsampled training images might relate closer to the images in the test split than another 10% subsample. Unfortunately, as you correctly mentioned below, providing quantitative variance numbers would require to re-run all analysis presented in the paper 9 times on the portion of ImageNet not considered in the initial 10% subsample. Specifically, we would need to fine-tune $9 \cdot 5=45$ DMs (5 DMs on 9 additional splits), for each of them generate images from 5 training epochs, resulting in $45 \cdot 5=225$ augmented datasets. For these, and the other modalities (2 baselines, 1 unconditioned DM, 4 prompt-based DMs, 1 retrieval), we would need to train 5 classifiers on each of the 9 splits to get proper variance estimates, resulting in a total of 1,485 classifier trainings. Unfortunately, we now have limited access to compute. Therefore, we sincerely apologize for being unable to re-run all of our experiments on the other 9 sub-partitions to provide these quantitative numbers.
>
> At the same time, we would expect performance numbers to differ only marginally, as the 10% random subsample follows the same image distribution as the full dataset. In addition, since the 10% subsplit is shared across all experiments, the performance of all models would be affected systematically in the same way. Assume for example, that a new 10% train-split would generalize better to the test split. Then all models would show increased performance: the 10% baseline, 20% ceiling, and all augmentation methods would benefit in the same way by the better training data they all share. Hence, the ordering of all methods according to their performance would not change. In conclusion, although single numbers might marginally vary, significant differences of performance and model ordering  is not expected by picking another 10% training split. This notion is in line with our results being consistent between 10% ImageNet and full ImageNet, which we elaborate on in the answer to your following question. We now have added an according discussion to paragraph “Limitations and future work” in the Discussion section.
>
>
> > Would the outcomes still be consistent with more data?
>
> This is a crucial question and after you raised it, we noted that we should have presented the according results better reflecting their importance. We verified on full 100% ImageNet that the best-performing fine-tuned DM outperforms the best frozen, prompt-conditioned DM, both improving over the baseline. Retrieval performed best in this setting, too, albeit by a much smaller amount. We could certainly have done a better job in visualizing this clearly and we now have added new Figure 1B highlighting that results obtained on 10% ImageNet are consistent with results on full ImageNet. Furthermore, we added new Figure 6 visualizing results obtained on full Caltech256. Moreover, we have improved clarity of the main text now clearly stating which results were obtained on which data (penultimate paragraph in Introduction, first paragraph in Results, improved paragraph “Augmenting full datasets” in Section 4.5, paragraph “Limitations and future work” in the Discussion) and added a discussion comparing performance numbers on full datasets to the data-scarce 10% ImageNet experiments (“Augmenting full datasets” in Section 4.5).
>
>
> > How much generated data would be needed to reach the performance of the proposed retrieval scenario?
>
> This is an interesting question we did not discuss in the previous version of the paper! From the scaling information depicted in Figure 5B we now have estimated that four to five times more synthetic data would be required to reach the same performance that is obtained by doubling training data through adding retrieved images. While a precise answer of this question is computationally infeasible, since it requires generating augmented data and training classifiers for many augmentation percentages, adding the estimate you asked for made the revised paragraph “Retrieval has better scaling behavior” (in Section “When does retrieval help?”) substantially more interesting – thank you very much!

---

> ### Author Response · Authors · 2023-10-19
>
> > The proposed retrieval method requires the whole training set of Stable Diffusion. I would expect this process to be computationally challenging. So it would be nice to compare compute requirements for all the groups of methods.
>
> We now have added discussion on compute requirements to the paper. Importantly, retrieval is substantially more efficient than generating images with diffusion models and does not require downloading the entire dataset. The key to make this process efficient is reusing a compute-optimized search index released by Laion which allows searching for those images which are most similar to a text prompt (Beaumont 2022). By reusing this index we can directly retrieve only those images that are required for augmentation from the web, without downloading the entire dataset. This is a substantial advantage, and we thank you for raising it.
>
> To provide quantitative numbers, we now have checked log files and file time-stamps estimating that retrieving images to augment the 10% ImageNet subsplit required 28.5 hours on a single machine. Beneficially, the retrieval code is very light-weight and just requires a 1.6TB SSD to store the index and 4 GB CPU RAM and – unlike diffusion models – does not require a GPU. For fine-tuning the diffusion models on 10% ImageNet, training times depended on the specific fine-tuning method, varying from 244 GPU hours to 720 GPU h (trained on 8 Nvidia V100 GPUs using distributed data parallel training), and synthesizing 390 images per ImageNet class required 867 GPU h on a Nvidia T4 GPU. In conclusion, reusing the existing search index renders retrieval to be very efficient, being 30-56x faster than DM-based methods and not requiring a GPU. We have now revised the paragraph  “Laion nearest neighbor retrieval.” and have added a new paragraph “Retrieval is computationally most efficient.” providing detailed compute requirements (both in new Section 3.1).
>
> Romain Beaumont. Clip retrieval: Easily compute clip embeddings and build a clip retrieval system with them.  https://github.com/rom1504/clip-retrieval , 2022.
>
>
> > - The conceptual comparison between GANs and diffusion models in the introduction is misleading. GANs can condition on text too. There the comparison is mainly between image generative vs text-to-image generative models, and the text should be revised accordingly.
> > - The first sentence of Section-3 does not read well.
> > - Figure-1 should be improved by mentioning what is measured in the figure.
> > - Taxonomy of methods in Figure-2B is not clear, and rather confusing. I don't have constructive feedback for this, sorry.
> > - It would be much better to place Figure-3 and Figure-4 before the experiments.
>
>
> Thank you for providing suggestions to further improve the text quality of our paper. We incorporated all of your suggestions into the main text, revised axis labels for Figure 1, added a legend to Figure 2B, and improved placement of all Figures and Tables throughout the paper.
>
>
>
> > It is significantly important to openly discuss and acknowledge potential data- and model-related biases in scenarios involving generating synthetic data to tackle specific tasks. Although the proposed method does not directly synthesize images, it utilizes uncurated web-data, which raises similar risks. I suggest the authors look at recent works in this field like [C], to develop a perspective, and add a Broader Impact Statement accordingly.
>
> We share your perspective on the importance of discussing potential data- and model-related biases, especially in the context of generating synthetic data and retrieving data from the web.Thank you for pointing out that this was missing in our paper. We now have added a new “Broader impact” section, extensively discussing potential societal and ethical impact and we have outlined how we mitigated potential risks.

---

> > ### Comment · Reviewer_JdHi · 2023-10-20
> >
> > Thanks for your reply, and adding so much more analysis to the manuscript.
> >
> > To summarize some of the important points in your response (please correct me if I'm wrong)
> > 1. The 10% of data selection was arbitrary
> > 1. Reporting variance is very costly.
> > 1. If more data is used, the performance gap between the retrieval-based method and the ImageNet training closes
> > 1. If ~4-5 times more data is used, then synthetic data is as effective as the retrieved ones from Lainon for trianing ImageNet classifier.

---

> > > ### Author Response · Authors · 2023-10-20
> > >
> > > > Thanks for your reply, and adding so much more analysis to the manuscript.
> > > >
> > > > To summarize some of the important points in your response (please correct me if I'm wrong)
> > > >
> > > > 1. The 10% of data selection was arbitrary
> > > > 2. Reporting variance is very costly.
> > > > 3. If more data is used, the performance gap between the retrieval-based method and the ImageNet training closes
> > > > 4. If ~4-5 times more data is used, then synthetic data is as effective as the retrieved ones from Lainon for trianing ImageNet classifier.
> > >
> > > 3. Retrieval performed +5.4% accuracy over the baseline on 10% ImageNet. You are right, adding more original training data, ie. using 100% ImageNet, reduces retrieval’s performance boost, resulting in a smaller yet not fully closed performance gap of +2.9% (blue vs. red bars in Figures 1A and 1B). Notably, on the full dataset, retrieval additionally outperforms the concurrent fine-tuned diffusion model of paper [B] you mentioned (gray bar in Figure 1B).
> > >
> > > 4. Using ~4-5 times the amount of data will make synthetic data competitive, which you correctly observed for the initial experiment adding images to double the augmented dataset (comparing performance of the synthetic data (blue) to the first datapoint (at 100% augmentation) of retrieval (orange) in Figure 5B). As more data is added augmenting with samples from both methods, the performance gap between augmentation with retrieved and synthetic images increases substantially (Figure 5B). Since the performance of classifiers trained on synthetic images saturates, it is unlikely that there will be sufficient data for the current generative methods to close the gap in a like-with-like comparison.
> > >
> > > We now have revised the paper to point this out more clearly (paragraph “Augmenting full datasets.” and paragraph “Retrieval has better scaling behavior.”, both in Section 4.5). Thank you very much for this discussion, as it helped us to further improve the paper.

---

### Author Response · Authors · 2023-10-19

We would like to thank **all reviewers** for their very helpful comments on our manuscript. We found it encouraging that the reviewers found our paper “well written and easy to follow” (Reviewer ixUD) “tackling an actively studied problem” (Reviewer JdHi) of “growing interest in this field” (JdHi), providing a “timely contribution” (Reviewer Kp2i) and a “thorough comparison” (JdHi). We are delighted that they “liked the straightforward approach” (ixUD) and “found the overall experimental setup clear” (Kp2i) and we are thrilled of their assessment of our suggested retrieval method being an “interesting baseline for future evaluation of data augmentation methods” (ixUD) “achiev[ing] superior performance” (JdHi) and the “discussion in Section 4 [(Results) being] overall interesting” (Kp2i). We addressed all the reviewers' concerns, included additional analyses in the revised manuscript, substantially revised writing, and provided exact information on the setting under which the claims in our paper were obtained. We believe that these have considerably improved the quality of our paper and we highly appreciate all reviewers' helpful feedback.

---

### Author Response · Authors · 2023-11-30
**Posted camera ready version of paper**

We want to thank all reviewers and the action editor once again for the helpful review process. We now have posted the camera ready version of our paper showing authors, affiliations and an Acknowledgements section.

---

### Decision · Action_Editor_Goau · 2023-11-09

**Recommendation:** Accept as is

**Comment:**

The authors made a substantial effort to address concerns raised by reviewers, and as a result reviewers feel that acceptance is warranted. I agree with them.

**Audience:**

Reviewers note that the submission makes a timely contribution given the growing interest in the problem of using synthetic data for training downstream models (JdHi, Kp2i), and that it makes an interesting baseline contribution (JdHi, ixUD).

**Claims And Evidence:**

Reviewers noted the submission's thorough experimental comparison of competing approaches (JdHi, ixUD). The following concerns have been addressed by the authors in their response:

- Positioning and comparison against He et al. (2023) and Azizi et al. (2023) (JdHi): The authors add a discussion about both He et al. (2023) and Azizi et al. (2023) in the manuscript and a comparison against Azizi et al. (2023).
- Concern that the approaches are not sufficiently well-trained (JdHi): The authors demonstrate that their experiments outperform results reported by Azizi et al. (2023).
- Concern that CLIP embeddings (which are trained on natural images) would cause the retrieval baseline to underperform on more specialized downstream tasks (ixUD): The authors acknowledge this as a limitation and discuss it in the limitations section of the manuscript. (This resolves the concern in that the authors do not make claims about strong performance in that setting.)
- Concern that Laion-5B overlaps with ImageNet, which would mean that the retrieval baseline "cheats" (Kp2i): The authors perform a pHash + manual inspection analysis showing that there is no significant overlap.
- Concern that the performance gap between the retrieval baseline and other approaches vanishes in larger data regimes (Kp2i): The authors acknowledge the phenomenon as expected given related work on the (in)significance of adding more data in large-data regimes. (This resolves the concern in that the authors do not make claims about a significant performance gap between the retrieval baseline and competing approaches in the large-data regime.)